# Dealing with Intimate Partner Violence and Family Violence in a Regional Centre of Western Australia: A Study of the Knowledge, Attitudes, and Practices of Local Social Workers

**DOI:** 10.3390/ijerph20095628

**Published:** 2023-04-24

**Authors:** Lindi Pelkowitz, Caroline Crossley, Heath Greville, Sandra C. Thompson

**Affiliations:** WA Centre for Rural Health, The University of Western Australia, Geraldton, WA 6530, Australia

**Keywords:** social workers, intimate partner violence, family violence, attitudes, knowledge, management, training, gender, safety plan

## Abstract

In the Midwest region of Western Australia, rates of intimate partner and family violence (IPV/FV) are high. We undertook research into social workers’ knowledge, attitudes, and skills as part of addressing this significant public health issue. Social workers come into contact with people experiencing IPV/FV in multiple settings, so their understandings and responses are critical to the prevention and interventions related to violence against women. The goal of the research was to determine the issues that the social workers in this region needed to be addressed that could assist in tackling the problem of IPV/FV. A questionnaire included open-ended questions to capture information on respondents’ profiles, knowledge, attitudes, practices, and education around IPV/FV, with 29 of 37 social workers working in the region responding. We also elicited respondents' recommendations related to training and service delivery. Despite working in many settings, most social workers had contact with people experiencing IPV/FV and had reasonable confidence and knowledge that showed an understanding of the complexity of FV, including why women stay in violent relationships. This paper identified social workers’ need for more education, including during their university training, resources, and service coordination to support best practice delivery of services to people affected by IPV/FV. Training to develop skills for conversations about IPV/FV with clients, around safety planning, and greater access to safe alternative accommodation for those leaving FV were identified priorities.

## 1. Introduction

In Australia, family violence is disturbingly common and contributes to multiple health and social problems, including poor mental and physical health, homelessness, and incarceration. Family Violence (FV) is defined as “violent or intimidating behaviors against a person, perpetrated by a family member including a current or previous spouse or domestic partner” [1]. A subset of FV is Intimate Partner Violence (IPV), defined by the World Health Organization (WHO) as “behavior by an intimate partner or ex-partner that causes physical, sexual or psychological harm, including physical aggression, sexual coercion, psychological abuse and controlling behaviors” [2]. Estimates of intimate partner violence presented in a study based on data from 161 countries showed that between 2000–2018, the prevalence of intimate partner violence among women aged 15–49 years ranged from 37% in “least developed countries” such as Polynesia and subregions of Africa, to 16–23% in counties such as Australia and subregions of Europe. This shows that violence against women is pervasive globally and is experienced by almost a third of all women worldwide [3].

The experience of FV or IPV can be lifelong with adverse consequences that often worsen over time, [4] including depression, anxiety, alcohol use disorders, early pregnancy loss, and increased rates of chronic pain disorders. Emotional health outcomes of IPV/FV can include low self-esteem, feelings of guilt or shame, and post-traumatic stress [5,6]. Survivors experience social consequences such as a lower employment rates and higher rates of homelessness, and are at higher risk of being victims of further violence [7].

While both men and women can be victims and perpetrators of intimate partner and family violence, the Australian Bureau of Statistics consistently indicates that these experiences are clearly gendered, with men being overwhelmingly the perpetrators. In addition, women are more likely than men to experience physical harm, express more fear, and experience subsequent psychological problems. It is for this reason, and to ensure that the questions were pertinent to the experience of social workers, that some questions in this survey pertain specifically to women.

Intimate partner homicide is the most common form of homicide in Australia and most victims are women [8]. It has been estimated that violence against women costs Australia $21.7 billion annually, with the largest component of this cost borne by victims, and governments spend $7.8 billion a year in health, administration, and social welfare costs toward addressing the costs of violence against women [9]. This is despite violence between intimate partners being recognized as among the most underreported of crimes [10].

Rates of FV are higher in rural areas, with the Midwest region of Western Australia (WA) experiencing the second highest rate of family violence in the state of WA. Local rates of assault by a family member are more than twice the state rate and more than three times the metropolitan rate [11]. However, FV in rural and remote communities has been given relatively little attention in Australia.

Geraldton is a coastal town in the Midwest region of the state of Western Australia, located about 425 km north of the state capital, Perth. It has a population of ~38,000, nearly 10% of whom are Aboriginal and with 76% of the population born in Australia [12]. In 2017, in response to ongoing high rates of FV, local service providers engaged with the community to catalyze efforts around the primary prevention of family violence, with the Community Respect Equality (CRE) Action Plan launched in September 2017 [13]. The CRE primary prevention initiative envisaged a collaborative effort to engage the whole community, including small businesses, sporting clubs, and service organizations. It was hoped this initiative would bring awareness and discussions about FV and the importance of respect for women and the role of gender equality out from the shadows into the sunlight within the Geraldton community.

The WA Centre for Rural Health (WACRH) committed as an organizational signatory to the CRE to both primary prevention of FV, including building other education and research activities to support its implementation, as well as contributing to better support for those impacted by FV. WACRH staff were aware of the occurrence of FV in the lives of many children and adults they interacted with as part of their work, and our interest increased by knowing that among our staff, many had been personally exposed to FV at some point in their life [14]. Among the many health professionals employed by WACRH are social workers, and it became clear in some of our work that they often received referrals from others in the health and social care system to help deal with consequences of FV.

Understanding the issues that individuals affected by FV are dealing with is critical if we are to ensure appropriate responses from frontline workers, who need to have the knowledge, skills, and attitudes to identify and manage FV. Social workers provide a broad range of services from within multiple agency settings and play a significant role in prevention, identification, and working with victim-survivors of IPV and FV. They are at the frontline of dealing with FV in crisis centers, emergency departments, after-hours services, and crisis accommodation, [15,16,17,18] including through supporting children in child protection who are affected by FV and supporting interventions within mental health services.

One of the roles of social workers is to “provide a significant contribution in the prevention and intervention of violence against women” [19]. Social workers are expected to have a client-focused holistic psychosocial and strengths-based approach, together with an array of theories, knowledge, and skills to address both broad and specific measures of prevention, assessment, crisis intervention, therapeutic intervention, case management, research, policy development, and advocacy related to IPV/FV [19,20]. However, we identified only two social work-specific studies in the literature prior to 2009 and four studies from 2009 onward that exclusively looked at social workers’ knowledge and practices related to IPV/FV.

In 2019, Labarre and co-authors found minimal research regarding how practitioners construct their understanding and views of perpetrators, victims, or service collaboration [21]. Yet better understanding social workers’ attitudes, beliefs, knowledge, and barriers to management can guide education and support for best-practice assessment and interventions around FV [15,21,22]. In the context of contemporaneous efforts to raise awareness of IPV/FV in Geraldton through the CRE, we decided to survey local social workers’ knowledge, attitudes, and management of FV. These beliefs and skills can either assist, support, or hamper those experiencing IPV/FV, thereby influencing the reporting and effectiveness of addressing IPV/FV. The survey aimed to contribute to understanding of management of FV and to explore whether there is a need for additional education or resources to assist and support social workers in best practice delivery of services for their clients affected by IPV/FV.

## 2. Materials and Methods

This was a survey that included multiple-choice, Likert-type responses and open-ended questions. It utilized a questionnaire modified from existing instruments, with questions adapted mainly from those in the PREMIS (Physician Readiness to Manage Intimate Partner Violence Survey) instrument, a validated 67-item questionnaire with good correlation with measured office IPV practices, including on repeat testing [23]. The original tool had sections that included respondent profiles, background information, IPV/FV knowledge, attitudes, and practice. The PREMIS tool was adapted by the research team over multiple iterations to suit the social work and local context and piloted with two social work colleagues. The adaption process involved several meetings of the research team to change wording to reflect currently used language, add questions to include information on updated information about family violence, and delete a section in order to make the survey less onerous timewise while still retaining the integrity of the instrument.

The final adapted survey consisted of demographic items, structured items, and optional open-response items, and included items on training, confidence, knowledge, attitudes, management, and challenges of working in FV. The full questionnaire is provided in Appendix A. The survey questions included Likert scales, true/false/don’t know items, and a range of multiple-choice options to determine knowledge. Open-response questions allowed participants to articulate in their own words their experiences and suggestions. The open-response questions asked respondents to describe the main challenges they experienced in discussing and managing IPV/FV with clients, any recommendations to improve the care of clients experiencing IPV/FV, interest in IPV/FV further education, and their preferred mode of delivery, including any specific areas or topics they would like to know more about.

### 2.1. Questionnaire Distribution

Considerable effort was expended to identify a complete list of social workers in Geraldton. All the social service organizations in the Midwest were contacted and asked to identify staff that were qualified social workers. A list was then compiled and displayed at social work network meetings so that social workers themselves could confirm or add names that had been missed. Some organizations were revisited when the final list was compiled to check the list. At the time of the survey, there were 39 social workers, and each one was hand-delivered the survey and personally invited by the lead author to participate in the study. The purpose of the study was explained, and the hard copy of the survey with a stamped addressed envelope was provided if participants were interested in completing the survey. All the social workers completed the consent form at the initial meeting; therefore, returned questionnaires were not linked to the respondents’ identity.

### 2.2. Data Analysis

The data from the survey were entered into Qualtrics and downloaded to the statistical analysis software, IBM SPSS V27.0, USA, for analysis. Tables were constructed to statistically analyze the data.

Demographic details (age, gender, area of SW practice, year of qualification, IPV/FV training) were sought from participants. Descriptive statistics were used to describe subgroups based on age, year of qualification, and training.

A thematic method of data analysis was undertaken, drawing on Brooks and King’s template analysis as a flexible form of thematic analysis involving preliminary coding, clustering, developing the initial template, modifying the template, defining the final template, using the template to interpret the data, and writing up. This suited the research as it is recognized as “a focusing technique that analyses data to identify tight generic themes originating withing the data, whilst simultaneously allowing for problem solving and theory building” [24].

### 2.3. Ethics Approval

The research project was approved by The University of Western Australia (UWA) (Project Number RA/4/20/5878). There were no financial or other incentives for participants for completing this questionnaire.

## 3. Results

### 3.1. Demographics

Thirty-nine social workers were invited to participate in the survey with 29 participants (26 female; 3 male) completing the surveys (74% response rate). The number of male participants was considered too small to report separately given issues of confidentiality and validity in any separate analysis. Around half (*n* = 15) of the participants were under 50 years of age, while almost 50% were over 50 years of age. Of the 28 respondents, 3 had graduated prior to 1990 and 8 had graduated since 2010, with 50% (*n* = 14) graduating prior to the year 2000 and 50% of participants (*n* = 14) graduating since 2000.

### 3.2. Exposure to IPV/FV Cases in the Field

Social workers encounter women and children living with domestic violence and abuse when working in a wide range of settings, including mental health agencies, hospitals, substance abuse treatment centers, government agencies such as the Department of Communities or the Department of Justice, and the education sector [5,15].

Participants in this survey listed 10 fields of social work practice. Five participants worked in Mental Health, five in Child Protection, five in Health/Hospitals, four in Family and Domestic Violence services, three in Justice/Correction services, three in Counseling services, two in the education sector, two in private practice, one in the disability sector, and one in community development. Two participants listed more than one area of employment e.g., Child Protection and Education. This indicates that social workers from a wide range of services, which included statutory agencies, non-government services, and private practice, participated in the survey.

The number of clients that social workers saw per week varied from 0–40. Almost 50% of the social workers (*n* = 14) said that they saw between 10–19 clients per week while ~20% (*n* = 6) saw 20–39 clients per week. Most (98%) reported that they had dealt with FV in their work with clients in the past 6 months, indicating that wherever they worked, the social workers encountered IPV/FV.

### 3.3. IPV/FV Training Received by Participants

Information on the nature of training that the social workers had received during coursework, student placements, and since graduation is shown in Figure 1 below.

While the majority (66%) of participants (*n* = 19) reported they had received formal education on IPV/FV in their social work coursework, 34% (*n* = 10) had received no formal education on IPV/FV in their social work coursework. Just over half (54%; *n* = 15) of participants reported they had received education or training on IPV/FV during their social work field placements, so nearly half had received no IPV/FV training while on placement.

Since graduation, nearly all (96%) of Geraldton social workers had received training in IPV/FV. Figure 1 shows the various modes of training they had received, with (79%; *n* = 23) reporting training within the workplace, and many being exposed to the issues through workshops or talks provided externally.

### 3.4. Confidence of Social Workers to Assess and Respond to IPV/FV

Most social workers (93%) were confident (always or almost always) to appropriately engage with a client who disclosed IPV/FV and to make appropriate referrals (89%) for a woman who had experienced IPV/FV. They were least confident (78%) about helping a client who had experienced IPV/FV to create a safety plan for herself and her children. However, this was at odds with many responses to the open-ended questions, which indicated that social workers felt a lack of confidence around asking clients questions about IPV/FV.

Of the various types of IPV/FV, social workers were most confident dealing with physical abuse (96%), social abuse (96%), and emotional abuse (93%). They were least confident in dealing with technological abuse (52%), spiritual abuse (56%), and reproductive control (52%).

When comparing the confidence of social workers, based on year in which they qualified, to assess and respond to the various types of IPV/FV (i.e., emotional, physical, sexual, economic, social, spiritual, reproductive control, technological, and elder abuse), those who qualified between 2000–2009 were the most confident group overall, with all of these participants (*n* = 6;100%) confident/very confident to respond to emotional, physical, sexual, economic, and social abuse. However, this group had the least confidence (46%) of all graduation cohorts when responding to reproductive control.

Interestingly, of the eight most recent social work graduates (2010–2019), six reported they learned about FDV at university and only three learned about FDV during field placements. The more recent graduates had less confidence compared with social workers who graduated between 1990–2009 when it came to engaging appropriately with a client who disclosed IPV/FV, and identifying history, signs, symptoms, and assessing the risk of a client who disclosed IPV/FV. Yet, these most recent graduates had the most confidence about documenting IPV/FV history, observations, and findings in the case notes compared with all the other groups of social workers.

### 3.5. Knowledge of IPV/FV

Although current research shows that being female is the single strongest risk factor for IPV/FV [25,26,27], less than half (41%) of social workers correctly answered “yes” to gender (see Table 1).

Survey responses to other knowledge questions about IPV/FV showed high levels of knowledge to all questions, except understanding that gender inequality is the underlying driver of IPV/FV (see Table 2).

Most (79%) participants were aware that alcohol and drugs are not the greatest predictor of the likelihood of IPV/FV. Most (93%) were aware that strangulation injuries indicate a high risk for IPV homicide, and that even if the child is not in immediate danger, practitioners have a duty of care to consider an instance of a child witnessing IPV/FV in terms of protecting the child (97%).

While the majority (83%) knew that women who have experienced IPV/FV are at greater risk of injury when they leave the relationship, 17% of the respondents either did not believe or did not know this information. Similarly, only 76% of the social workers were aware that pregnant women are at higher risk of experiencing IPV/FV, with 24% unaware or not believing this. Seventeen percent were unsure or did not believe that reasons for concern about IPV/FV should be included in a woman’s case notes if she does not disclose the violence. All respondents (100%) were aware that allowing a partner to be present during the consultation of a woman who has experienced IPV/FV does not ensure her safety.

Most (83%; *n* = 24) identified that perpetrators use violence as a means of controlling their partner, with a smaller proportion seeing a family history of abuse (*n* = 5) as common.

### 3.6. Overall Attitudes, Opinions, and Practice of Social Workers Regarding IPV/FV

All the social workers (100%) in this survey believed that women who remain in an IPV/FV relationship are not responsible for the violence, with 82% understanding that women experiencing IPV/FV cannot readily leave the relationship even if they want to. Almost half (48%) believed that women experiencing IPV/FV can make appropriate choices about how to handle their situation, and that being supportive of a woman’s choice to remain in a violent relationship does not condone the abuse (83%). Social workers (97%) did not believe that IPV/FV is a private matter or that as a health professional they should not interfere by asking about it if the patient does not directly disclose. Respondents (93%) overwhelmingly disagreed that if a woman who is experiencing IPV/FV does not acknowledge the abuse, there is very little help they can offer.

Almost 14% of the respondents agreed that they were concerned about their legal obligations if a client were to disclose IPV/FV, and 10% were worried about their own safety when working with a woman who has disclosed IPV/FV. Only 7% of the social workers agreed that women living with IPV/FV often misuse alcohol or other drugs.

### 3.7. Management of IPV/FV

The survey questioned respondents about their time and work set-up in terms of responding to clients with IPV/FV (Table 3). While the majority (76%) of respondents felt they had adequate private space to provide support to women experiencing IPV/FV, almost 20% did not have adequate private space. Most (86%) had the time to ask about IPV/FV, although two responded that they did not ask about IPV/FV because of having little or no experience or training in IPV/FV, while 14% selected a neutral response to this question, possibly reflecting different responses contingent upon circumstances. Only 73% felt they had the support of colleagues in dealing with disclosure of IPV/FV. Three quarters (72%) felt they could match interventions to a woman’s readiness to change.

Most social workers reported that they always or nearly always enquired about IPV/FV when children had unexplained signs or symptoms (89%), and when clients had physical injuries (85%). They reported lower likelihood of nearly always/always enquiring about IPV/FV when working with clients with alcohol/drug misuse (76%), if there were mental health concerns (72%), family breakdown (72%), homelessness (71%), and social concerns (68%), or when clients presented with financial difficulties. A minority (32–36%) reported consistently enquiring about IPV/FV when clients presented with eating disorders, sleeping disorders, or unemployment.

### 3.8. Actions Taken by Social Workers When Clients Have Been Identified as Experiencing IPV/FV

All but two respondents had identified IPV/FV in the past 6 months. The three most common responses (from nearly 90% of participants) were to offer those they identified as experiencing IPV/FV relevant information in the form of pamphlets or phone numbers, provide counseling to women about options they may have, and to make referrals to local services.

Approximately 75% (*n* = 20) had conducted a safety assessment for the woman (and her children where applicable), and 60% (*n* = 16) had helped to develop a personal safety plan. Seven respondents reported they had assisted clients in other ways, such as by providing reports and court support, one driving a woman who was escaping to a metropolitan city 5 hours away, one providing support to a man experiencing FV, and one reporting working with offenders.

### 3.9. Referrals to Other Services

Only 62% of respondents (*n* = 18) felt they had adequate knowledge of referral resources in the community, including shelters or support groups, for women who have experienced IPV/FV, while 38% (*n* = 11) did not feel they had adequate knowledge of referral resources. Respondents listed multiple professionals, services, and organizations they had referred clients to in the last year. Almost all (97%) listed the agency that was funded for family violence services and that managed the women’s refuge and other accommodation services. Other agencies to which referrals were made included legal services (16), health services such as Geraldton Regional Aboriginal Medical Services (5) and general practitioners (5), hospital (4), the Community Mental Health Service (3), and counselling services provided by a non-government social services organization (8). There were also referrals to the Police (8), Child Protection (4), and Centrelink (3), which provides social security payments and services to Australians, as well as a small number of referrals to other health service providers such as homelessness and support services, including telephone support for mental health and those experiencing IPV/FV.

### 3.10. Analysis of Open-Ended Comments

#### 3.10.1. Challenges Social Workers Perceive in Discussing and Managing IPV and FV

All respondents provided information to the open-response question, “What are the main challenges you have in discussing and managing IPV/FV with clients?” with the following themes emerging from the analysis.

#### 3.10.2. Fear and a Lack of Confidence

Despite most (93%) social workers always or almost always confident in the Likert responses for knowledge about IPV/FV, a third described feelings of fear (*n* = 5) and a lack of confidence (*n* = 10) working in the field of IPV/FV. This pertained to a lack of confidence asking clients questions about IPV/FV, making comments related to offending or angering the client and getting them offside: “not sure how to ask questions” or how to “start a conversation if (I) suspect IPV/FV but it is not disclosed”, “I fear that my judgement is wrong and that’ll plant a seed in the client and destroy their relationship”. Four of the three social workers working in the field of mental health feared that asking about IPV/FV could disrupt the therapeutic relationship with the client. A child protection worker commented that “parents are worried that children will be removed” and that it was challenging “when parents did not understand the impact of IPV/FV on children”. One participant noted that “working with children in (these) family contexts, (and) when disclosures from children can be denied, (one is) more inclined to refer to Child Protection Services”. Participants working with perpetrators noted it was challenging work, needing care not to collude or be coercive with the perpetrator.

Fear for “client safety” was another issue because minimal housing options and limited resources available to victim-survivors make the role for social workers assisting clients experiencing IPV/FV challenging and difficult.

#### 3.10.3. Lack of Appropriate Accommodation and Housing for Victim-Survivors of IPV/FV

Participants consistently commented on there being minimal housing or safe accommodation to which victim-survivors could be referred. The refuge was reported as often full, with women and children usually only permitted to stay in the refuge for three weeks. Comments included “I can’t provide suitable safe alternative accommodation (for women wanting to leave perpetrators of FV), and (the) fit for some of the local population experiencing IPV/FV is not optimal”, “There is no culturally appropriate accommodation”, and “A lot of clients do not feel comfortable in the local refuge”. One participant commented that there were limited shelters for men suffering from IPV.

#### 3.10.4. Limited Support Services and Lack of Resources

Seven respondents noted there were minimal local support services in the region, which resulted in challenges with referrals to adequate services and income support, and that resources were even more limited in (remote/rural) communities outside of Geraldton and noted “being unable to ensure ongoing support”. Three of these participants specifically mentioned a lack of resources for men experiencing IPV and a need for improved perpetrator accountability services.

In the quantitative section of the survey, over one third (36%) of social workers were unaware of all the services that catered for clients experiencing IPV/FV, a finding reiterated in some open-ended responses where social workers described that it was challenging to make referrals when they did not know all the services that provide support for clients experiencing IPV/FV.

#### 3.10.5. Lack of Cultural Safety

Social workers were concerned about the lack of culturally appropriate accommodation options for women leaving violent relationships as finding “safe places for our clients to go that are culturally appropriate is really difficult”. Additional challenges reported when working with Aboriginal clients were different gender and cultural issues, that IPV/FV was normalized by some, and complex socio-economic issues that impede behavioral change.

#### 3.10.6. Logistics and Organizational Support

Some social workers expressed that it can be challenging to get victim-survivors away from their partners long enough to talk to them, and described difficulties with getting the correct address and contact details to locate clients. They recognized that the cost of attending some services created an additional challenge, with some clients not returning post-disclosure.

A small number felt there was a passive acceptance from colleagues around the issue of IPV/FV, and a lack of consistency of support to engage with a client possibly experiencing IPV/FV, with IPV/FV seen as “not core business” within some organizations. Social workers in these environments reported being discouraged from engaging in a conversation with clients even if they suspected IPV/FV. A shortage of staff and a lack of private space/rooms in some organizations to have the conversation with clients about IPV/FV impeded responding.

### 3.11. Data Analysis of Education Questions

#### 3.11.1. Social Work Education

Most (*n* = 20; 83%) of the respondents wanted further education regarding IPV/FV, with one third of the participants interested in more education around early detection and intervention for IPV/FV. As one social worker wrote, “Training! IPV/FV is complex. I have found that social workers whilst meaning well, often have a shallow understanding of the issue and impact on all the members including perpetrators”. Upskilling of social workers, intensive modules, more available training, appropriate material, and workshops for professionals to develop increased understanding of IPV/FV were recommendations from recent graduates as well as seasoned social workers. The preferred format for education was in the form of workshops (*n* = 16; 70%), with 45% (*n* = 9) specifically requesting that workshops be skills-based. Some of those interested in education wanted online training (*n* = 9; 45%), whereas for 7 (35%), their preferred mode of delivery was a university module.

#### 3.11.2. Community Education

Participants (*n* = 6) recommended “ongoing preventive education and raising community awareness”. Comments from the open-ended questions included “More education, increased understanding of issues in the greater community (culture shift) so it is more openly talked about” to “prevent negative views and assumptions of those experiencing IPV/FV”. Open engagement with the conversations about FV, such as the impacts of FV on future families, and the importance of encouraging healthy relationships, healthy communication, and respect was recommended to develop “better understanding and responding across different sections–primary health/mental health, and community-based agencies”.

#### 3.11.3. Policies and Procedures

Participants suggested that workplaces need to provide clear guidelines, policies, and protocols to help people understand what to do should IPV/FV come up.

### 3.12. Social Workers’ Recommendations for Improving the Care of Clients’ Experiencing IPV/FV in Social Work Practice

Four themes emerged from recommendations made by the social workers: better service coordination, more accommodation options, more community resources, and additional education, as has been reported above.

Participants recommended linked collaboration and networking between different agencies to demonstrate “a flexible and responsive service tailored to clients’ individual needs” and “agencies networking collaboratively to ensure clients do not ‘fall through the cracks’”. “Social work requires in most cases a referral from other agencies to a specific service for a client. I would wish for an easy-to-access first starting point for social work services”. “Men and women, perpetrator and victim services are usually very separate. This is for good reasons but isn’t always helpful as the division is not that clean and the problems are enmeshed”. The need for ongoing support to deal with the complex issues related to relationship breakdown related to IPV/FV was also noted: “services are often short term when evidence suggests that work in this area requires long term intervention and support with all parties to bring about change”. The respondents noted the need for client-friendly services and more culturally appropriate resources for women as part of creating more flexible and responsive services tailored to clients’ individual needs.

There is a significant shortage of housing in the region. The need to increase the number of safe houses and “having at least one culturally appropriate refuge” in Geraldton was noted, along with a clearer and quicker pathway to access Centrelink payments/emergency relief for women who leave their partner.

Among other issues raised were the ongoing need for perpetrator accountability programs for men focused on behavior change and life skills training and counseling for women.

## 4. Discussion

This snapshot of social workers’ knowledge, attitudes, and management of Intimate Partner Violence/Family Violence (IPV/FV) in a regional town in Western Australia in 2019 adds to evidence that social workers regularly come into contact with women and children living with IPV/FV in multiple workplace settings [5].

A key finding of this research is the lack of IPV/FV education these social workers had received at university. Between a third to half of survey participants graduated with either no education about IPV/FV in their university coursework or during field placements. All participants accessed IPV/FV education post-graduation in the form of attending a talk, attending a workshop or course. “While a dearth of knowledge exists…early studies found that social workers themselves reported not having the skills to deal with domestic violence cases…and the need for specific training in this area” [5]. A literature search indicates similar results have been found internationally and have been an issue for over 30 years [5,28,29].

This finding implies that the modern social work curriculum must ensure students have the training and knowledge necessary to engage in practice that promotes social and economic justice for survivors of IPV/FV and their children [30,31]. Unfortunately, no recent surveys of IPV/FV education and social work curricula have been published; thus, the extent to which the current Master in Social Work and Bachelor of Social Work programs offer IPV/FV-specific training and education to social work students is unknown. While placement with agencies as part of social work training could expose students to issues of IPV/FV, it is also possible that students never obtain any practical learning in an agency that is experienced in dealing with the problems of IPV/FV; therefore, more consideration for how social work students could gain practical relevant quality experience during training is needed.

Participants also identified the need after qualifying for ongoing education and skill-building in the workplace. The majority (80%) of participants wanted this, including identifying some specific areas, such as safety planning. In addition, continuing education opportunities to build skills in IPV/FV for all social workers who are in practice is needed. Offering such training more broadly across a regional area to ensure social workers without an evident case load dealing with IPV/FV could help build a common understanding of underpinning risks, responses, and support pathways.

Given the challenges of working in a complex and contested space related to conflict, participants were keen for their workplaces to have policies and procedures to provide clear guidelines on managing IPV/FV.

The survey results indicate that most participants were aware of IPV/FV perpetrators using violence as a means of controlling their partners, that strangulation injuries indicate a high risk for intimate partner homicide, and that even if a child is not in immediate danger, practitioners have a duty of care to consider an instance of a child witnessing IPV/FV in terms of child protection. However, there were considerable gaps in knowledge, since between 17% and 24% did not appreciate the higher risks for pregnant women, the greater risk of injury for women when they leave the relationship, and that alcohol and drugs are not the greatest predictor of the likelihood of IPV/FV. Understanding the importance of gender inequality as an underlying driver of IPV/FV to be addressed in order to prevent violence against women was recognized by less than half of the respondents. Although social workers generally work at the treatment end of health care, increased understanding of what drives IPV/FV could enhance their awareness of coercive control, gaslighting, and other controlling behaviors, as well as making them greater allies in primary prevention of IPV/FV.

In addition, many (38%) of the social workers felt they did not have adequate knowledge of referral resources in the community. The open-ended questions enabled the participants to elaborate on this as a challenge and make recommendations to address resource issues. Those participants who were confident about what resources exist in the community expressed concern over the lack of housing options and the need for a more coordinated response and long-term interventions to support victim-survivors. Recent research has emphasized the critical importance of tangible resources for women during this post-separation period [32,33,34]. Women who are trying to escape abusive partners need services to meet their needs. These services include immediate crisis intervention such as food and shelter, longer-term assistance in overcoming the emotional or psychological impact of domestic violence on themselves and their children, and assistance related to economic security and housing stability.

Participants shared several challenges while interacting with IPV/FV victim-survivors. Almost a third of the participants lacked confidence and experienced fears when working with victims of IPV/FV. This fear was not in terms of their own safety but related to asking victims questions about IPV/FV. This is concerning because research indicates that “victims want the issue of domestic violence to be proactively and routinely raised by health professionals in order to make it easier to disclose abuse” [35]. In addition to personal attitudes, social workers’ perceptions of their readiness or confidence in their own ability to respond to IPV/FV can assist or impede effective clinical responses to victims [16]. As well as possessing the required skills, individuals need to feel confident in their abilities to use those skills and execute tasks in the workplace [36] in order to provide care and support for victims of IPV/FV.

Social workers identified the need for more resourcing and housing, including culturally appropriate refuges, and wanted a more coordinated and long-term investment of services to cater for victim-survivors of IPV/FV. This suggests that there is a significant amount of scope for the improvement of service delivery and support needed in the field of IPV/FV in Geraldton. Social workers also identified that services in small communities in more remote/rural areas serviced by Geraldton do not have resident IPV/FV support services. At best, they may have some external support in the form of telehealth or a fortnightly/monthly/ bi-annual visit from a social service agency situated in a regional town several hours away. Social workers recognized the need for more resources to support those experiencing IPV/FV outside the regional city.

It is possible that the limited resources available to victim-survivors can also act as a barrier to social workers’ confidence and ability to assist victim-survivors. Some participants found it challenging to manage the therapeutic relationship when aware of inadequate resources or support for FV victim-survivors. This is consistent with previous research [37]. They also found it challenging working with mothers who suffer FV when the mothers are not willing to disclose. This includes women not recognizing their abuse, fearing their children will be removed, and/or prioritizing the abusive relationship above adverse impacts on their children. As one participant explained, when “…you have suspicions, and then you have to take it further…then you have a family who don’t trust you anymore…so your intervention is hampered.” These factors and the lack of knowledge about resources and perceived lack of resources add to the low confidence of some social workers working with clients who may be experiencing IPV/FV.

Given that the pathway of leaving an IPV/FV relationship is usually long, there are ethical and material barriers that impact social workers’ confidence as well as their fears around being able to adequately assist clients who they suspect or know are experiencing IPV or FV.

Past research on attitudes shows that social workers blame the victims of IPV/FV for being in a domestic violence relationship. The current study indicated that social workers in Geraldton do not have the same negative attitudes toward IPV/FV victims that were part of previous social work practice, where social workers blamed the victims for their situation, and many considered issues of low self-esteem as the primary reason why victims remained in abusive relationships [36].

One of the limitations of this research is that it did not include coercive control as a specific area of knowledge. Another limitation was the size of the survey with 29 participants, despite the very respectable (74%) response rate. Extending the survey to a larger number of participants would only be possible by expanding the geographic area, although our intention was to look at the issues for social workers in the regional city. However, meaningful comparison of responses between years of education and experience since qualifying with the knowledge, attitudes, and management of IPV/FV would require an expanded survey.

## 5. Conclusions

Social workers in Geraldton are exposed to IPV/FV in multiple workplace settings and would like additional education and skills-based training in IPV/FV. While this survey showed that most social workers have current knowledge of IPV/FV and understand the complexities of IPV/FV, the survey highlights that there is a significant number of social workers who lack skills and confidence and do not have familiarity with key elements of current knowledge about IPV/FV. Our findings suggest that IPV/FV is often not adequately covered during university social work training, with learning about IPV/FV generally gained after graduation, through attending talks or workshops. Social workers were also keen to see more community education about IPV/FV in terms of primary prevention of FDV. Social workers in this survey highlighted the need for more local resources, such as housing options, and required more information about referral service options as well as improved collaboration between agencies in the region.

## Figures and Tables

**Figure 1 ijerph-20-05628-f001:**
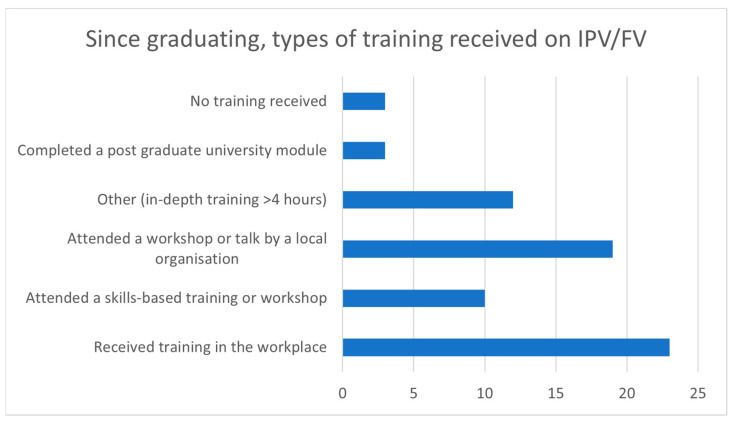
Postgraduate Training in IPV/FV.

**Table 1 ijerph-20-05628-t001:** Knowledge of strongest single risk factor for IPV/FV.

Knowledge of the Strongest Single Risk Factor for IPV/FV	Social Workers Response	% Correct Response
Age < 30 years	0%	
Partner abuses alcohol/drugs	(*n* = 5) 17%	
Gender-Female	(*n* = 12) 41%	✓
Family history of abuse	(*n* = 12) 41%	
Don’t know	0%	

**Table 2 ijerph-20-05628-t002:** Knowledge regarding IPV/FV statements of risk and causation (*n* = 29).

Statements	Answer Should Be	% Answered Accurately
Alcohol consumption is the greatest single predictor of the likelihood of IPV/FV	False	79
Women who have experienced IPV/FV are at a greater risk of injury when they leave the relationship	True	83
Gender inequality is the underlying driver of violence against women	True	41
Allowing a partner to be present during the consultation of a woman who has experienced IPV/FV ensures her safety	False	100
Pregnant women are at higher risk of experiencing intimate partner violence	True	76
Strangulation injuries indicate a high risk for IPV homicide	True	93
Reasons for concern about IPV/FV should not be included in a woman’s case notes if she does not disclose the violence	False	83
Even if the child is not in immediate danger, practitioners have a duty of care to consider an instance of a child witnessing IPV/FV in terms of child protection.	True	97

**Table 3 ijerph-20-05628-t003:** Current practice when seeing clients who may be experiencing IPV/FV or who have disclosed abuse.

Survey Questions	% Agree or Strongly Agree	% Neutral	% Disagree or Strongly Disagree
There is adequate private space for me to provide support for women who are experiencing or have experienced IPV/FV	76	7	17
I generally do not have the time to ask about IPV/FV	3	11	86
I tend not to ask about IPV/FV because I have little or no experience or training in this subject	7	14	79
I am reluctant to ask about IPV/FV because I might offend the client or make matters worse	11	7	82
I have the support of colleagues in dealing with disclosure of IPV/FV	73	10	17
I can match interventions to a woman’s readiness to change	72	21	7

## Data Availability

The data analyzed during the current study are not publicly available due to small participant numbers and protection of confidentiality.

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
