# Peer review of "Dealing with Intimate Partner Violence and Family Violence in a Regional Centre of Western Australia: A Study of the Knowledge, Attitudes, and Practices of Local Social Workers"

_ijerph, 2023, doi:10.3390/ijerph20095628_

Round 1

Reviewer 1 Report

This article presents findings on social workers' knowledge and attitudes around domestic and family violence in a regional area of Australia. The researchers used mixed methods and found that social workers want more education, training and support when working with victim/survivors of domestic and family violence. As there is limited information on the practice experiences of social workers in rural Australia, and even less on their experiences with domestic and family violence, this article offers important insights.

I found the article was extremely readable, and the data clearly outlined and explained. I thought using the term IPV/FV was out of step with the most common terminology in Australia, which is domestic and family violence. However, this may have been done due to the survey instrument used, which did use the terminology of IPV. It could be worth considering how to ensure the findings are consistent with the terminology used by the AASW for example, as I think these findings would be very relevant for their training and other workshops.

Overall, I found this article presented the research clearly and highlighted an underresearched area. Well done to the authors!

Author Response

Attached please find the revised responses to Reviewer 1

Reviewer 2 Report

This is a very interesting topic. Indeed the authors present a simple and unambiguous narration of the situation in their location of study. The problem of the study was aptly identified, as the objectives and justification for the study were clearly stated in the manuscript.

Besides, the authors selected and applied appropriate and simple methods and techniques in identifying the respondents, collecting the required data and making informed analyses.

However, some issues need further attention by the authors.

·      1)   The combination of the two different types of violence somewhat derailed the researchers from investigating deeply and specifically into one of the phenomena. It becomes very difficult to understand the suitability of the instruments used to collect the required data. Particularly on whether they are generic enough to accommodate these two separate phenomena. I suggest that the authors consider using FV as the focus of this study, since FV includes IPV.

·        2) The entire study is premised on a biased assumption that men are never victims of IPV/FV or those men are the only perpetrators. I tend to agree with this lopsided orientation. The study appears to have omitted the men side of the story. It is misleading to place the story that IPV/FV is about women and children alone. The question is: where are the men?

     3) If the intention of the authors is to leave out the menfolk, then the title of this Manuscript has to change to specifically capture the narrow perspective (women and children) being narrated in this manuscript. In other words, the title of the Manuscript does not align with the content, which is women and children centered, instead of family centered where all members of the family are included, without excluding men.

·        4) Table 1 needs to be introduced after the narrative.

5) Throughout this Manuscript, I did not see any attempt by the authors to make a comparative analysis between and among variables. It will be more interesting to see a more detailed comparative analysis of the IPV/FV situation in this study.

Author Response

Please see attached response to Reviewer 2.

Reviewer 3 Report

In the abstract, the authors should make the goal more explicit and make the exposition more coherent. They can indicate how many participants were involved and better specify what findings emerge from the study and their relevance to the context.

The introduction defines the situation of family violence in Australia and then rightly refers to the World Health Organization. Perhaps the authors could further expand the introduction by reporting some data about the phenomena of violence in Australia and compare them with those of other countries of the world.

In "method and materials", the authors begin the paragraph by defining the study as a "mexed methods study", however I do not think this definition is adequate. In fact, it seems to me that the analysis of the closed responses consisted only in counting frequencies. The authors did not perform correlations, statistical tests for significance or a regression analysis between quantitative variables (things that would have proved complicated anyway given the low sample size).

In the same paragraph, the authors say that "The PREMIS tool was adapted by the research team over multiple iterations..." but they should provide more information about the tool adaptation process.

In "distribution of the questionnaire", the authors say that "Considerable effort was expended to identify a complete list of social workers", but do not clarify what these efforts consisted of. The sample selection procedure should be made more explicit.

In the discussions, the authors could implement the text by comparing their results with other international sources and articles, underlining whether or not these results differ from what has already been found in the literature.

For example they might look at:

Messing, J.T., & Thaller, J. (2015). Intimate partner violence risk assessment: A primer for social workers. The British Journal of Social Work, 45(6), 1804-1820.

Esposito, C., Di Napoli, I., Esposito, C., Carnevale, S., & Arcidiacono, C. (2020). Violence against women: a not in my back yard (NIMBY) phenomenon. Violence and gender, 7(4), 150-157.

Yechezkel, R., & Ayalon, L. (2013). Social workers' attitudes towards intimate partner abuse in younger vs. older women. Journal of family violence, 28, 381-391.

Goldblatt, H., Buchbinder, E., Eisikovits, Z., & Arizon-Mesinger, I. (2009). Between the professional and the private: The meaning of working with intimate partner violence in social workers' private lives. Violence against women, 15(3), 362-384.

Author Response

Please see response comments to Reviewer 3.

Round 2

Reviewer 2 Report

Having gone through the revised version, I have come to the conclusion that the manuscript is acceptable to be published in its current form and version.